Comparison of the performances of six empirical mass transfer-based reference evapotranspiration estimation models in semi-arid conditions

http://orcid.org/0000-0001-8970-7333 Usta Selçuk susta@yyu.edu.tr
Van Vocational School Department of Construction Technology, Yüzüncü Yil (Centennial) University , Van , Turkey
Yapıcı Sercan
Electronic publication date: 2024 Nov 27
Publication date: 2024
Volume: 12
Electronic Location ID: e18549
Received 2024 Jul 19; Accepted 2024 Oct 28
Copyright: © 2024 Usta
Copyright year: 2024
Copyright holder: Usta
License: This is an open access article distributed under the terms of the Creative Commons Attribution License, which permits unrestricted use, distribution, reproduction and adaptation in any medium and for any purpose provided that it is properly attributed. For attribution, the original author(s), title, publication source (PeerJ) and either DOI or URL of the article must be cited.
License URL: https://creativecommons.org/licenses/by/4.0/

Keywords: Calibration, Estimation model, Mass transfer, Penman–Monteith, Reference evapotranspiration, Reliability analysis

Funding: The author received no funding for this work.

==============================
Background

Accurately measured or estimated reference evapotranspiration (ETo) data are needed to properly manage water resources and prioritise their future uses. ETo can be most accurately measured using lysimeter systems. However, high installation and operating costs, as well as difficult and time-consuming measurement processes limit the use of these systems. Therefore, the approach of estimating ETo by empirical models is more preferred and widely used. However, since those models are well in accordance with the climatic and environmental traits of the region in which they were developed, their reliability must be examined if they are utilised in distinctive regions. This study aims to test the usability of mass transfer-based Dalton, Rohwer, Penman, Romanenko, WMO and Mahringer models in Van Lake microclimate conditions and to calibrate them in compatible with local conditions.

Methods

Firstly, the original equations of these models were tested using 9 years of daily climate data measured between 2012 and 2020. Then, the models were calibrated using the same data and their modified equations were created. The original and modified equations of the models were also tested with the 2021 and 2022 current climate data. Modified equations have been created using the Microsoft Excel program solver add-on, which is based on linear regression. The daily average ETo values estimated using the six mass transfer-based models were compared with the daily average ETo values calculated using the standard FAO-56 PM equation. The statistical approaches of the mean absolute error (MAE), mean absolute percentage error (MAPE), root mean square error (RMSE), Nash–Sutcliffe Efficiency (NSE), and determination coefficient (R2) were used as comparison criterion.

Results

The best and worst performing models in the original equations were Mahringer (MAE = 0.70 mm day−1, MAPE = 15.86%, RMSE = 0.87 mm day−1, NSE = 0.81, R2 = 0.94) and Penman (MAE = 1.84 mm day−1, MAPE = 33.68%, RMSE = 2.39 mm day−1, NSE = −0.49, R2 = 0.91), respectively, whereas in the modified equations Dalton (MAE = 0.29 mm day−1, MAPE = 7.51%, RMSE = 0.33 mm day−1, NSE = 0.97, R2 = 0.97) and WMO (MAE = 0.36 mm day−1, MAPE = 8.89%, RMSE = 0.43 mm day−1, NSE = 0.95, R2 = 0.97). The RMSE errors of the daily average ETo values estimated using the modified equations were generally below the acceptable error limit (RMSE < 0.50 mm day−1). It has been concluded that the modified equations of the six mass transfer-based models can be used as alternatives to the FAO-56 PM equation under the Van Lake microclimate conditions (NSE > 0.75), while the original equations—except for those of Mahringer (NSE = 0.81), WMO (NSE = 0.79), and Romanenko (NSE = 0.76)—cannot be used.

Introduction

Evapotranspiration (ET) is defined as the total mass of water vapour losses occurring through evaporation (E) from soil and water surfaces and transpiration (T) from crops. ET, which is one of the most difficult elements of the hydrological cycle to estimate, constitutes the primary data for many studies, such as determining the irrigation water requirements of crops and preparing irrigation schedules, designing, constructing, and operating irrigation and drainage systems, ponds, and dams, determining the amount of precipitation infiltrating groundwater, monitoring drought, and estimating the safe yield of groundwater basins (Abudu et al., 2011; Jing et al., 2019). The most accurate and reliable actual evapotranspiration (ETa) values are measured by lysimeter and Eddy covariance systems. However, the use of these systems is limited due to high installation and operating costs as well as complex and time-consuming measurement processes. Therefore, the method of estimating crop evapotranspiration (ETc) by correcting the reference evapotranspiration (ETo) with the crop coefficient (Kc) is more preferred than lysimeter and Eddy covariance systems (Şarlak & Bağçacı, 2020). In this method, ETo is defined as evapotranspiration realised an extensive surface of green, well-watered grass of uniform height, actively growing and completely shading the ground. Kc is a coefficient that varies depending on the crop types and the length of the crop-growing season (Allen et al., 1998).

Penman-Monteith is the most widely used method for estimating ETo. This method was developed in 1948 and improved by adding some constants over time. It was adapted to the reference grass crop by the Food and Agriculture Organization of the United Nations (FAO) in 1998 and was made available under the name of Penman–Monteith (PM) equation, FAO-56 modification (Penman, 1948; Allen et al., 1998). In FAO-56 PM equation (Eq. (1)); ETo is reference evapotranspiration (mm day−1), Rn is net radiation (MJ m−2 day−1), Δ is slope of saturation vapour pressure curve (kPa °C−1), γ is psychrometric constant (kPa °C−1), U2 is wind velocity at 2 m above the ground surface (m s−1), es and ea are saturation and actual vapour pressures (kPa), respectively, and G is soil heat flux (MJ m−2 day−1).

(1) ETo=0.408×Δ×(Rn−G)+γ×(900T+273)×U2×(es−ea)Δ+γ×(1+0.34×U2).

The FAO-56 PM method is combination-based and has been validated for use in different climatic conditions without the need for local calibration. Therefore, it has been accepted as universal standard method (Sentelhas, Gillespie & Santos, 2010; Berti et al., 2014). In this method, ETo can be accurately estimated using all parameters: air temperature, relative humidity, solar radiation, wind velocity, and soil heat flux. If any of the parameters are missing, the method cannot be used. Among these parameters, the hard to measure is solar radiation. Solar radiation can vary significantly over short time intervals, depending on the cloud density in the atmosphere. There are large differences in the amounts of solar radiation measured during daylight and nighttime conditions. For these reasons, it is necessary to measure solar radiation at short intervals of 15–30 min throughout the day to determine the daily total and calculate the average at the end of the day. Additionally, the outer glass dome of the pyranometer used to measure solar radiation needs to be cleaned periodically. These factors increase labor and time usage (El-Sebaii et al., 2010). The climate parameters that constitute the input variables of the FAO-56 PM equation are measured by meteorology ground observation stations. However, since these stations are not widespread enough all over the world and are mostly located in urban centres, many of these parameters cannot be measured continuously and regularly in rural areas. There may be problems in data supply. Another problem is that the costs of the devices used in measurement processes are high and their calibration is difficult and time-consuming (Ener Ruşen, 2017). Therefore, it becomes necessary to use empirical ETo estimation models based on air temperature (Thornthwaite, 1948; Hamon, 1961; Blaney & Criddle, 1962; Schendel, 1967; Hargreaves, 1975) and solar radiation (Makkink, 1957; Turc, 1961; Jensen & Haise, 1963; Priestley & Taylor, 1972; McGuinness & Bordne, 1972; Doorenbos & Pruitt, 1977; Hargreaves & Samani, 1985; Abtew, 1996) which require the use of fewer meteorological parameters as input variables in rural areas (Tabari, Grismer & Trajkovic, 2013; Al-Shibli et al., 2021). Most of these models were developed based on linear regression between ETo and some meteorological parameters. However, there is a non-linear relationship between ETo and meteorological parameters, which makes it difficult to estimate ETo using linear regression-based models. Additionally, since these models are compatible with the climatic and environmental characteristics of the regions in which they were developed, it is necessary to test their reliability and calibrate them, if needed, when they are used in different regions (Landeras, Ortiz-Barredo & Lopez, 2008; Izadifar & Elshorbagy, 2010).

Unlike temperature-based and radiation-based estimation models, ETo can also be estimated based on the principle of determining the total mass of water vapour transferred from soil and plant surfaces to the atmosphere. For this purpose, many empirical estimation models based on the atmospheric vapour pressure deficit (VPD) between the surfaces of soil and crops and the air surrounding these surfaces have been developed (Dalton, 1802; Trabert, 1896; Meyer, 1926; Rohwer, 1931; Penman, 1948; Albrecht, 1950; Romanenko, 1961; WMO, 1966; Mahringer, 1970). VPD is determined by subtracting the actual vapour pressure (ea) of the air from the saturation vapour pressure (es) just above the soil and crop surfaces (VPD = es − ea). Mass transfer-based models are based on the principle that water vapour formed as a result of evaporation is transferred from a very dense environment to a less dense environment by eddy movement. While the temperature of the water and the velocity of its molecules increase in parallel with the increase in air temperature, the surface tension decreases and therefore evaporation becomes easier. In order for evaporation to continue, water vapour must move away from the water surface by diffusion and convection. This is possible with the air movement. Wind plays an important role in continuing evaporation by increasing the dispersion through turbulent diffusion in the vertical direction. It is known that evaporation increases by 1–3% depending on a 10% increase in wind velocity (Terzi, 2011). Therefore, unlike temperature-based estimation models, ETo estimations with higher accuracy can be made with mass transfer-based models that taken into account wind velocity as well as temperature. With that being said, the fact that solar radiation cannot be measured continuously and regularly at some stations around the world limits the usability of radiation-based estimation models (Ener Ruşen, 2017). This situation increases the preferability of mass transfer-based models, which require the use of fewer meteorological parameters that can be easily measured or available compared to temperature-based and radiation-based models (Valipour, Gholami Sefidkouhi & Raeini-Sarjaz, 2017).

Many studies have been conducted to test the reliability and usability levels of mass transfer-based empirical ETo estimation models in different climatic and environmental conditions. Islam et al. (2020) compared the daily average ETo values estimated by the original and calibrated equations of some mass transfer-based estimation models with the ETo values obtained using the standard FAO-56 PM method in a study conducted in semi-arid Abha region, Saudi Arabia. In this study, the ETo values with the highest accuracy were estimated by Saif and Albrecht models, while Meyer and Trabert models had the worst estimation performance. Valipour, Gholami Sefidkouhi & Raeini-Sarjaz (2017) determined the usability levels of eleven mass transfer-based ETo estimation models in Iran conditions and calibrated them separately for thirty-one cities with different climatic characteristics. The Penman model performed better than other models in fifteen cities before calibration and twenty-one cities after calibration. Rezaei, Valipour & Valipour (2016) compared the daily average ETo values estimated using some mass transfer-based models with the values determined by the FAO-56 PM method. In this study conducted for twenty-three cities in Iran, the ETo values with the highest accuracy were estimated with the Albrecht model. Similarly, in another study conducted in humid regions of Iran, among the ten mass transfer-based estimation models evaluated by Tabari, Grismer & Trajkovic (2013), Romanenko was the model with the best estimation performance. Djaman et al. (2015) expressed that Trabert, Mahringer and Albrecht models, which were calibrated in compatible with local climatic and environmental characteristics, were the best performing models in the semi-arid Senegal River valley. Muniandy, Yusop & Askari (2016) reported that Penman was the model with the highest estimation performance among some mass transfer-based models evaluated in Malaysian conditions. Bormann (2011) compared the Dalton, Trabert, Meyer, WMO and Mahringer models based on mass transfer with some models based on temperature and radiation under Germany conditions. Significant differences were observed between the performances of the evaluated models. Rim (2000) stated that the Dalton model performed better than the Penman Model in Arizona, United States of America. Singh & Xu (1997) demonstrated that the evaporation values estimated by Meyer, Dalton and Rohwer models tested in four different climatic conditions of Canada were reasonable agreement with the observed actual values. It has been observed that almost all of the mass transfer-based models evaluated within the scope of these studies are very sensitive to changes in air temperature, wind velocity and relative humidity parameters. Moreover, these models have shown the worst performance in many studies where they have been compared with models based on air temperature and radiation (Rácz, Nagy & Dobos, 2013; Farzanpour et al., 2019; Proutsos et al., 2023). For this reason, it is necessary to determine the reliability levels by comparing them with the ETo values obtained using the lysimeter or the standard FAO-56 PM method and to make calibrations compatible with local climatic and environmental conditions (Bogawski & Bednorz, 2014).

It was observed that the temperature-based and radiation-based models were mostly evaluated in the studies realised to test the usability of empirical ETo estimation models in Van city conditions. Usta & Gençoğlan (2019) created ETo estimation models based on multiple linear regression using temperature and relative humidity as independent variables. They reported that these methods can be used as an alternative if all the data required for the standard FAO-56 PM method cannot be measured or supplied. In other studies conducted by Usta et al. (2019) and Uzunlar, Öz & Diş (2022), the original equations of the temperature-based Blaney & Criddle and radiation-based Hargreaves & Samani models were tested under local conditions and modified equations were created. They found that the modified equations compatible with local conditions performed better than the original equations. Unlike previous studies, this study aims to evaluate the usability and reliability of mass transfer-based ETo estimation models—Dalton (1802), Rohwer (1931), Penman (1948), Romanenko (1961), WMO (1966), and Mahringer (1970)—in Van Lake microclimate conditions, and to calibrate them for compatibility with local conditions using the linear regression-based solver add-on of Microsoft Excel program.

Materials and Methods

Van Lake is located on the high plateau in the Eastern Anatolia Region of Turkey. It is the largest soda lake on Earth and one of the ten largest closed-basin salty lakes in the world. The total surface area of Van Lake closed basin is 15,496 km2, which includes surface area of Van Lake (3,582 km2) and land area of basin (11,914 km2) (Degens et al., 1984). Van is located east of Van Lake, between 37°43′–39°26′ north latitudes and 42°40′–44°30′ east longitudes. The surface area of Van is 19,062 km2 (Fig. 1). It is the sixth largest city of Turkey in terms of surface area. The altitude of this city within the borders of the Van Lake closed basin is 1,726 m and the annual average air temperature and humidity are 9.50 °C and 58.67%, respectively. It is one of the most sun-drenched cities in Turkey with an annual average daily sunshine duration of 7.90 h day−1 and a solar radiation intensity of 15.32 MJ m−2 day−1. The total annual precipitation is 392.70 mm. Precipitation decreases significantly during the May–August period when the daily maximum air temperature ranges between 24.00–28.50 °C. In this period, the monthly total precipitation ranging between 5.80–18.40 mm is insufficient to satisfy crop water consumption and irrigation becomes compulsory (Turkish State Meteorological Service, 2022).

Figure 1 Geographical location of Van and Van Lake on the Turkey map.

Drawing credit: Selçuk Usta.

According to the reports of the Intergovernmental Panel on Climate Change (IPCC), it has been stated that serious climate change effects are observed in Turkey’s Mediterranean, Aegean, Eastern and Central Anatolia Regions (IPCC, 2013). Therefore, Van city within the boundaries of Eastern Anatolia Region, which is under the risk of drought according to IPCC reports, has been chosen as the study area. In addition, Van city, located on the shores of Van Lake, offers the opportunity to produce an estimate of reference evapotranspiration using the FAO-56 PM equation and therefore allows a comparison between different estimation models. The majority of the population’s livelihood in Van city is based on agriculture. Due to the drought experienced in recent years in the Eastern Anatolia Region, where Van is located, there is a greater need for irrigation than ever before to prevent yield and quality losses in agricultural production. It is predicted that yield and quality in agricultural production in Turkey will decrease due to the drought problem, which is seen as one of the greatest challenges of our age, accordingly leading to issues in food supply and the economy (Arslan & Ergül, 2014; Uzunlar, Öz & Diş, 2022). In addition, the agricultural sector consumes approximately 70% of the world’s water resources. Almost all of this ratio consists of water used for irrigation purposes (Küçük, Parlakçı Doğan & Aydoğdu, 2022). Sustainable use of water resources can be ensured by irrigation within irrigation schedules sensitive to real-time crop water consumption (Jensen & Allen, 2016). Accurately measured or estimated ETo data is needed for the preparation of irrigation schedules.

The original equations of the Dalton, Rohwer, Penman, Romanenko, WMO and Mahringer models were evaluated using 9 years of daily climate data measured between 2012 and 2020. These models were calibrated using the 9-year average values of the daily climate data and their modified equations were created. The original and modified equations of the six mass transfer-based models were also tested using the 2021 and 2022 current climate data. In this direction, daily average air temperature (T), relative humidity (RH), wind velocity at 2 m above the ground surface (U2) and solar radiation (Rs) data supplied from Edremit ground observation station numbered 17,172 of Van Regional Directorate of Meteorology are given in Fig. 2. The altitude of this station is 1,669 m and it is located at 38°28′ north latitude and 43°20′ east longitude (Turkish State Meteorological Service, 2022). Some statistical parameters of the climate data are shown in Table 1. In the table can be seen the average, minimum and maximum values of climate data, standard deviation and skewness. Standard deviation and skewness values of T, RH, U2 and Rs data, which have non-normal distributions, were determined using the Microsoft Excel program. A skewness value between −1 and +1 is excellent, while −2 to +2 is generally acceptable. Values beyond −2 and +2 suggest substantial non-normal (Hair et al., 2022). Since the skewness values are excellent for T, RH, and Rs, and acceptable for U2, the data have not been transformed into a normally distribution (Table 1). Similarly, in many previous studies where some empirical ETo estimation methods were evaluated and calibrated, daily average T, RH, Rs, and U2 data with non-normal distributions were used without being transformed into a normal distribution (Tabari, Grismer & Trajkovic, 2013; Djaman et al., 2016, 2017; Valipour, 2017; Proutsos et al., 2023). The study was conducted for the April–October period, considering the growing seasons of agricultural crops such as apple, walnut, sugar beet, rye, barley, and wheat, which are predominantly cultivated in Van (TAGEM, 2017).

Figure 2 Daily average air temperature (T), relative humidity (RH), wind velocity (U2) and solar radiation (Rs) values.

Each line on the graphs represents the 9-year averages of daily average T, RH, U2, and Rs values measured between 2012 and 2020 (black lines), as well as the daily average T, RH, U2, and Rs values measured in 2021 (red lines) and 2022 (blue lines).

Table 1 Daily statistical parameters of the climate data.

Period	Data	Maximum	Minimum	Average	Standard deviation	Skewness	
2012–2020	T (°C)	23.26	4.44	16.42	5.12	−0.32	
RH (%)	75.39	39.83	53.98	8.90	0.49	
Rs (MJ m−2 day−1)	24.88	11.88	20.15	3.53	−0.32	
U2 (m s−1)	6.30	1.48	2.65	0.77	1.41	
2021	T (°C)	25.00	5.56	15.97	5.02	−0.38	
RH (%)	84.80	29.60	57.20	10.24	0.53	
Rs (MJ m−2 day−1)	26.22	11.50	19.90	4.05	−0.51	
U2 (m s−1)	7.91	1.34	2.88	1.10	1.33	
2022	T (°C)	27.12	5.23	17.09	5.08	−0.55	
RH (%)	79.30	22.50	46.23	10.62	0.67	
Rs (MJ m−2 day−1)	26.84	7.56	20.69	4.38	−0.94	
U2 (m s−1)	8.63	1.20	2.97	1.03	1.48	
Note:

T is daily average air temperature (°C). RH is daily average relative humidity (%). Rs is daily average solar radiation (MJ m−2 day−1), and U2 is daily average wind velocity at 2 m above the ground surface (m s−1).

The original equations of the Dalton, Rohwer, Penman, Romanenko, WMO, and Mahringer models, as well as the modified equations aimed to be created after the calibration of these original equations, are given in Table 2. The daily average ETo values calculated using the standard FAO-56 PM equation (Eq. (1)) were accepted the actual ETo values within the scope of the study. The components of the six mass transfer-based ETo estimation models were determined using the Eqs. (2)–(6). The components of FAO-56 PM equation were determined using the Irrigation and Drainage Publication No. 56 prepared by the FAO (Allen et al., 1998).

Table 2 Original and modified equations of the mass transfer-based ETo estimation models.

Models	References	Original equations	Modified equations	
Dalton	Dalton (1802)	ETo=(0.3648+0.07223×U2)×VPD	ETo=a×[(0.3648+0.07223×U2)×VPD]+b	
Rohwer	Rohwer (1931)	ETo=0.44×(1+0.27×U2)×VPD	ETo=a×[0.44×(1+0.27×U2)×VPD]+b	
Penman	Penman (1948)	ETo=0.35×(1+0.98/100×U2)×VPD	ETo=a×[0.35×(1+0.98/100×U2)×VPD]+b	
Romanenko	Romanenko (1961)	ETo=0.00006×(T+25)2×(100−RH)	ETo=a×[0.00006×(T+25)2×(100−RH)]+b	
WMO	WMO (1966)	ETo=(0.1298+0.0934×U2)×VPD	ETo=a×[(0.1298+0.0934×U2)×VPD]+b	
Mahringer	Mahringer (1970)	ETo=(0.15072×3.60×U2)×VPD	ETo=a×[(0.15072×3.60×U2)×VPD]+b	
Note:

“a” and “b” are calibration coefficients. ETo is daily average reference evapotranspiration (mm day−1). VPD is daily average atmospheric vapour pressure deficit. VPD is in hPa in all the models except for the Rohwer and Penman, where VPD is in mmHg. U2 is daily average wind velocity at 2 m above the ground surface. U2 is in m s−1 in all the models except for the Penman, where U2 is in miles day−1. RH is daily average relative humidity (%), and T is daily average air temperature (°C).

(2) VPD=es−ea

(3) esmax=0.6108×exp⁡(17.27×TmaxTmax+237.3)

(4) esmin=0.6108×exp(17.27×TminTmin+237.3)

(5) es=esmax+esmin2

(6) ea=[(esmin×RHmax100)+(esmax×RHmin100)]2.

In the above equations; VPD is atmospheric vapour pressure deficit. VPD is in hPa in all the models except for the Rohwer and Penman, where es, ea, esmax, and esmin are in mmHg. es and ea are saturation and actual vapour pressures, respectively. esmax is saturation vapour pressure at the maximum air temperature, and esmin is saturation vapour pressure at the minimum air temperature. es, ea, esmax, and esmin are in hPa in all the models except for the Rohwer and Penman, where es, ea, esmax, and esmin are in mmHg. Tmax and Tmin are maximum and minimum air temperatures, respectively (°C). RHmax and RHmin are maximum and minimum relative humidity values (%) (Allen et al., 1998).

The modified equations of the six mass transfer-based models were created by assigning calibration coefficients “a” and “b” to the original equations. The optimal values of the calibration coefficients were determined using the Microsoft Excel program solver add-on, which is part of the Microsoft Excel program’s simulation analysis tools command set. The solver works with a group of cells that have a direct or indirect relationship with the formula in the target cell. The solver determined calibration coefficients based on linear regression between the daily average actual ETo values (ETo (FAO-56 PM)) calculated using the FAO 56-PM equation and the daily average ETo values (ETo (MTBM)) estimated using the original equations of the six mass transfer-based models (MTBM) (Eq. (7)). The actual ETo (FAO-56 PM) values are considered the dependent variable, while the estimated ETo (MTBM) values are considered the independent variables. The constant “a” (slope) and “b” (intercept) defined as calibration coefficients. The calibrated equations must have slope (a) close to unity while intersept (b) should be near to zero for best result. In order to estimate calibrated coefficient a (slope) multiply the slope of a regression line by inversing the slope in order to make the slope of equation closer to unity. Also to get “b” (intercept) closer to zero opposite sign value of intercept was added for new regression equation (Islam et al., 2020). The modified equations aimed to be created after the calibration of the original equations, are given in Table 2.

(7) ETo(FAO−56PM)=a×(ETo(MTBM))+b

In the calibration studies performed using the Microsoft Excel program, firstly the input variables (es, ea, VPD, Δ, γ, Rn, G) of the six mass transfer-based models and the FAO-56 PM equation were calculated using the 9-year average values of the daily T, RH, U2 and Rs climate data measured between 2012 and 2020. Then, actual daily average ETo values were determined using the FAO-56 PM. Nine-year average values of the daily climate data and actual ETo values were entered into the Microsoft Excel program and formula definitions were made for the modified equations of the six mass transfer-based models. The calibration coefficients “a” and “b” in these equations were assigned to the “changing cells” section of the Excel solver and the value “1” was initially entered in these coefficients. The sum of the squares of the differences between the ETo values, estimated by entering the value “1” into the calibration coefficients and the actual ETo values was calculated and assigned to the “target cell” in the solver. Finally, the solver was run and the optimal “a” and “b” coefficients that minimize this sum were obtained. As a result of these calibration processes, performed separately for each model, modified equations compatible with the Van Lake microclimate conditions were created (Fylstra et al., 1998; Briones & Escola, 2019).

The daily ETo values estimated using the original and modified equations of the mass transfer-based models were compared with the actual daily ETo values determined using the FAO-56 PM equation. The statistical approaches of the mean absolute error (Eq. (8)), mean absolute percentage error (Eq. (9)), root mean square error (Eq. (10)), and Nash–Sutcliffe Efficiency value (Eq. (11)) were used as comparison criterion. The accuracy of the estimated ETo values was considered “excellent” if MAPE < 10%, “good” if MAPE = 10–20%, “reasonable” if MAPE = 20–50% and “inaccurate” if MAPE > 50%. In addition, according to the evaluation based on NSE, the accuracy of the estimated ETo values was considered “good” if NSE > 0.75, “satisfying” if NSE = 0.36–0.75 and “less satisfactory” if NSE < 0.36 (Nash & Sutcliffe, 1970; De Myttenaere et al., 2016; Maiseli, 2019; Lufi, Ery & Rispiningtati, 2020). Regression analyses were performed to reveal the level of statistical relationship between actual and estimated ETo values (Eq. (12)).

(8) MAE=1n∑i=1n|Xi−Yi|.

(9) MAPE=1n∑i=1n|Xi−YiXi|×100

(10) RMSE=1n∑i=1n(Xi−Yi)2

(11) NSE=1−∑i=1n(Xi−Yi)2∑i=1n(Xi−X^)2

(12) R2=[∑i=1n(Xi−X^)×(Yi−Y^)]2∑i=1n(Xi−X^)2×∑i=1n(Yi−Y^)2.

In the above equations; MAE is mean absolute error (mm day−1), MAPE is mean absolute percentage error (%), RMSE is root mean square error (mm day−1), NSE is Nash–Sutcliffe Efficiency value, Xi and Yi are actual and estimated ETo values (mm day−1), respectively, X^ and Y^ are averages of actual and estimated ETo values (mm day−1), respectively, R2 is determination coefficient, and n is number of observations.

Results

The calibration coefficients (a, b) of Dalton, Rohwer, Penman, Romanenko, WMO, and Mahringer models were determined via the linear regression-based solver add-on of Microsoft Excel program using the 9-year average values of the daily climate data measured between 2012 and 2020. The modified equations created as a result of correcting the original equations with calibration coefficients (a, b) are given in Table 3.

Table 3 Calibration coefficients and modified equations of the mass transfer-based models.

Models	a	b	Modified equations	
Dalton	0.5457	1.3459	ETo=0.5457×[(0.3648+0.07223×U2)×VPD]+1.3459	
Rohwer	0.5286	1.3849	ETo=0.5286×[0.44×(1+0.27×U2)×VPD]+1.3849	
Penman	0.4555	1.5061	ETo=0.4555×[0.35×(1+0.98/100×U2)×VPD]+1.5061	
Romanenko	0.5220	1.6315	ETo=0.5220×[0.00006×(T+25)2×(100−RH)]+1.6315	
WMO	0.7402	1.5715	ETo=0.7402×[(0.1298+0.09340×U2)VPD]+1.5715	
Mahringer	0.6372	1.4336	ETo=0.6372×[(0.15072×3.60×U2)×VPD]+1.4336	
Note:

“a” and “b” are calibration coefficients. ETo is daily average reference evapotranspiration (mm day−1). VPD is daily average atmospheric vapour pressure deficit. VPD is in hPa in all the models except for the Rohwer and Penman, where VPD is in mmHg. U2 is daily average wind velocity at 2 m above the ground surface. U2 is in m s−1 in all the models except for the Penman, where U2 is in miles day−1. RH is daily average relative humidity (%), and T is daily average air temperature (°C).

The daily average actual ETo values determined using the FAO-56 PM equation varied between 1.29–10.14 mm day−1 during the period between 2012 and 2020 (Figs. 3–6). For the same period, the daily ETo values estimated by the original equations of the Dalton, Rohwer, Penman, Romanenko, WMO, and Mahringer ranged between 1.07–13.87, 1.05–14.48, 1.08–16.77, 1.17–11.20, 0.49–11.65, and 0.73–12.58 mm day−1, respectively. The daily ETo values estimated by the modified equations of the same models ranged from 1.77–10.38, 1.80–10.57, 1.83–10.91, 1.56–9.59, 1.86–11.03, and 1.79–10.62 mm day−1, respectively. The estimated ETo values with the highest and lowest statistical correlation to the daily actual ETo values were obtained using the Mahringer (R2 = 0.94) and Penman (R2 = 0.91), respectively, in the estimates made using the original equations. The estimates using the modified equations were obtained with Dalton (R2 = 0.97) and Romanenko (R2 = 0.95), respectively.

Figure 3 Daily average actual and estimated reference evapotranspiration (ETo) values for Dalton, Rohwer and Penman models (2012–2020).

The black lines on the graphs represent the actual ETo values calculated using the FAO-56 PM equation. The blue and red lines represent the ETo values estimated using the original and modified equations of the models, respectively.

Figure 4 Daily average actual and estimated reference evapotranspiration (ETo) values for Romanenko, WMO and Mahringer models (2012–2020).

The black lines on the graphs represent the actual ETo values calculated using the FAO-56 PM equation. The blue and red lines represent the ETo values estimated using the original and modified equations of the models, respectively.

Figure 5 Statistical analysis of the relationship between actual and estimated reference evapotranspiration (ETo) values for Dalton, Rohwer and Penman models (2012–2020).

The black points on the graphs represent the actual ETo values calculated using the FAO-56 PM equation. The blue and red points represent the ETo values estimated using the original and modified equations of the models, respectively.

Figure 6 Statistical analysis of the relationship between actual and estimated reference evapotranspiration (ETo) values for Romanenko, WMO and Mahringer models (2012–2020).

The black points on the graphs represent the actual ETo values calculated using the FAO-56 PM equation. The blue and red points represent the ETo values estimated using the original and modified equations of the models, respectively.

The monthly average actual ETo values determined for the period 2012–2020 reached the highest levels in July (5.87 mm day−1) and August (5.58 mm day−1), and decreased to the lowest levels in April (2.49 mm day−1) and October (2.56 mm day−1). They were realized as 3.57, 4.91 and 4.36 mm day−1 in May, June and September, respectively. The seasonal average ETo for the April–October period of 2012–2022 was obtained as 4.20 mm day−1. The seasonal averages nearest to actual seasonal ETo (4.20 mm day−1), estimated using the original equations of the Mahringer (4.35 mm day−1), Romanenko (4.82 mm day−1), WMO (3.56 mm day−1), Dalton (5.24 mm day−1), Rohwer (5.33 mm day−1) and Penman (5.92 mm day−1), respectively. In contrast, estimates made with the modified equations yielded values (4.20 mm day−1) equal to the actual seasonal average ETo. (4.19 mm day−1) (Tables 4, 5).

Table 4 Monthly average ETo (mm day−1) values estimated using the original models (2012–2020).

Months	FAO-56 PM	Dalton	WMO	Mahringer	Penman	Rohwer	Romanenko	
Apr	2.49	2.14	1.41	1.75	2.36	2.16	2.31	
May	3.57	3.39	2.35	2.83	3.89	3.47	3.29	
Jun	4.91	5.71	3.94	4.78	6.54	5.84	5.17	
Jul	5.87	8.11	5.53	6.75	9.20	8.27	7.01	
Aug	5.58	8.17	5.49	6.76	9.16	8.30	7.17	
Sep	4.36	6.04	4.05	4.99	6.77	6.13	5.71	
Oct	2.56	3.10	2.13	2.57	3.54	3.17	3.11	
Avg.	4.20	5.24	3.56	4.35	5.92	5.33	4.82	

Table 5 Monthly average ETo (mm day−1) values estimated using the modified models (2012–2020).

Months	FAO-56 PM	Dalton	WMO	Mahringer	Penman	Rohwer	Romanenko	
Apr	2.49	2.51	2.61	2.55	2.58	2.53	2.48	
May	3.57	3.19	3.31	3.24	3.28	3.22	3.15	
Jun	4.91	4.46	4.49	4.48	4.49	4.47	4.44	
Jul	5.87	5.77	5.66	5.73	5.70	5.76	5.71	
Aug	5.58	5.82	5.63	5.74	5.68	5.77	5.82	
Sep	4.36	4.64	4.57	4.62	4.59	4.63	4.81	
Oct	2.56	3.04	3.15	3.07	3.12	3.06	3.03	
Avg.	4.20	4.20	4.20	4.20	4.20	4.20	4.20	

Using the original equations of the six mass transfer-based models, the daily ETo values with the lowest and highest errors in the estimates made for the period between 2012 and 2020 were estimated using the Mahringer and Penman models, respectively. The MAE, MAPE, RMSE, and NSE values of the Mahringer, which has the best-estimating performance, were calculated as 0.70 mm day−1, 15.86%, 0.87 mm day−1, and 0.81, respectively. The same values were determined as 1.84 mm day−1, 33.68%, 2.39 mm day−1, and −0.49 for the Penman, which has the worst-estimating performance. The model showing the nearest performance to Mahringer was Romanenko (MAE = 0.79 mm day−1, MAPE = 16.47%, RMSE = 0.99 mm day−1, NSE = 0.76). The original equations of the WMO (MAE = 0.79 mm day−1, MAPE = 20.46%, RMSE = 0.90 mm day−1, NSE = 0.79), Dalton (MAE = 1.24 mm day−1, MAPE = 23.26%, RMSE = 1.58 mm day−1, NSE = 0.35), and Rohwer (MAE = 1.32 mm day−1, MAPE = 24.63%, RMSE = 1.70 mm day−1, NSE = 0.25) showed lower performance than the Mahringer and Romanenko. The accuracy ranking of the original equations from best to worst according to their performance for daily average ETo estimates is as follows: Mahringer > Romanenko > WMO > Dalton > Rohwer > Penman. The daily ETo values estimated by Mahringer and Romanenko models have a good level of accuracy (MAPE = 10–20%), while the daily average ETo values estimated by other models have reasonable accuracy (MAPE = 20–50%). According to the NSE performances, Romanenko, WMO and Mahringer have a good level of accuracy (NSE > 0.75), while Dalton, Rohwer and Penman have less satisfactory accuracy (NSE < 0.36) (Table 6).

Table 6 Performances of the original models in estimating daily ETo (2012–2020).

Models	MAE
(mm day−1)	MAPE
(%)	RMSE
(mm day−1)	NSE	Accuracy	
MAPE	NSE	
Dalton	1.24	23.26	1.58	0.35	Reasonable	Less satisfactory	
Rohwer	1.32	24.63	1.70	0.25	Reasonable	Less satisfactory	
Penman	1.84	33.68	2.39	−0.49	Reasonable	Less satisfactory	
Romanenko	0.79	16.47	0.99	0.76	Good	Good	
WMO	0.79	20.46	0.90	0.79	Reasonable	Good	
Mahringer	0.70	15.86	0.87	0.81	Good	Good	
Note:

Mean absolute error (MAE), mean absolute percentage error (MAPE), root mean square error (RMSE) and, Nash–Sutcliffe Efficiency (NSE) express the deviation between the daily average actual ETo values calculated using the FAO-56 PM equation and the daily average ETo values estimated using the original equations of the Dalton, Rohwer, Penman, Romanenko, WMO and Mahringer models.

The modified equations compatible with Van Lake microclimate conditions showed very similar performances. The MAE, MAPE, RMSE, and NSE values of the daily average ETo values estimated for the period between 2012 and 2020 using the modified equations varied between 0.29–0.36 mm day−1, 7.51–8.89%, 0.33–0.44 mm day−1, and 0.95–0.97, respectively. The daily average ETo values with the lowest and highest errors were estimated using the Dalton and WMO models, respectively. The MAE, MAPE, RMSE, and NSE values of the Dalton, which has the best-estimating performance, were calculated as 0.29 mm day−1, 7.51%, 0.33 mm day−1, and 0.97, respectively. The same errors were determined as 0.36 mm day−1, 8.89%, 0.43 mm day−1, and 0.95 for the WMO, which has the worst-estimating performance. The accuracy ranking of the modified equations from best to worst according to their performance for daily average ETo estimates is as follows: Dalton > Rohwer > Mahringer > Penman > Romanenko > WMO. The daily average ETo values estimated by modified equations have a excellent level of accuracy (MAPE < 10%) according to MAPE performances and good level of accuracy (NSE > 0.75) according to NSE performances. The nearest values to the actual daily average ETo values were estimated using the modified equations (Table 7).

Table 7 Performances of the modified models in estimating daily ETo (2012–2020).

Models	MAE
(mm day−1)	MAPE
(%)	RMSE
(mm day−1)	NSE	Accuracy	
MAPE	NSE	
Dalton	0.29	7.51	0.33	0.97	Excellent	Good	
Rohwer	0.29	7.65	0.34	0.97	Excellent	Good	
Penman	0.33	8.42	0.40	0.96	Excellent	Good	
Romanenko	0.36	8.86	0.44	0.95	Excellent	Good	
WMO	0.36	8.89	0.43	0.95	Excellent	Good	
Mahringer	0.31	7.85	0.36	0.97	Excellent	Good	
Note:

Mean absolute error (MAE), mean absolute percentage error (MAPE), root mean square error (RMSE) and, Nash–Sutcliffe Efficiency (NSE) express the deviation between the daily average actual ETo values calculated using the FAO-56 PM equation and the daily average ETo values estimated using the original equations of the Dalton, Rohwer, Penman, Romanenko, WMO and Mahringer models.

According to the results of the calibration processes, the model with the most increased estimation performance was Penman. The performance of this model, which has the worst-estimating performance with an accuracy of 66.32% (MAPE = 33.68%) in daily average ETo estimations made with the original equations, increased by 38.08% to 91.58% (MAPE = 8.42%) after calibration. The MAE (1.84 mm day−1), MAPE (33.68%) and RMSE (2.39 mm day−1) errors of daily average ETo values estimated using the original equation of the Penman model decreased by 82.07%, 75.00% and 83.26% to 0.33 mm day−1, 8.42% and 0.40 mm day−1, respectively. The performance of the Mahringer model, which has the best-estimating performance with an accuracy of 84.14% (MAPE = 15.86%) in daily average ETo estimations made with the original equations, increased by 9.57% to 92.15% (MAPE = 7.85%) after calibration. The MAE (0.70 mm day−1), MAPE (15.86%) and RMSE (0.87 mm day−1) errors of daily average ETo values estimated using the original equation of the Mahringer model decreased by 55.71%, 50.50% and 58.62% to 0.31 mm day−1, 7.85% and 0.36 mm day−1, respectively. Using the original equations of the Rohwer, Dalton, WMO, and Romanenko models, the daily average ETo values were estimated with an accuracy of 75.37% (MAPE = 24.63%), 76.74% (MAPE = 23.26%), 79.54% (MAPE = 20.46%), and 83.53% (MAPE = 16.47%), respectively. These accuracy rates increased by 22.52%, 20.65%, 14.55% and 9.11% after calibration, respectively. Using the modified equations of the Rohwer, Dalton, WMO, and Romanenko models, daily average ETo values with an accuracy of 92.35% (MAPE = 7.65%), 92.59% (MAPE = 7.51%), 91.11% (MAPE = 8.89%) and 91.14% (MAPE = 8.86%) were estimated, respectively.

The original and modified equations of the six mass transfer-based models were also tested with the current climate data of 2021 and 2022. The daily actual ETo values calculated using the FAO-56 PM for the April–October periods of 2021 and 2022 varied in the ranges 1.76–7.52 and 1.38–9.08 mm day−1, respectively (Figs. 7, 8). Similarly, the daily ETo values estimated using the original equations of the Dalton, Rohwer, Penman, Romanenko, WMO and Mahringer ranged between 0.83–12.89, 0.85–13.74, 0.95–16.86, 0.87–9.22, 0.58–10.74, and 0.70–11.35 mm day−1 in 2021, and 0.85–17.21, 0.84–18.61, 0.87–23.41, 1.00–10.90, 0.51–14.79, and 0.65–15.12 mm day−1 in 2022, respectively. The daily ETo values estimated using the modified equations of the same models varied between 1.80–8.33, 1.83–8.65, 1.94–9.18, 1.49–7.22, 2.00–9.52, and 1.88–8.67 mm day−1 in 2021, and 1.81–10.74, 1.83–11.22, 1.90–12.17, 1.59–8.37, 1.95–12.52, and 1.85–11.07 mm day−1 in 2022, respectively.

Figure 7 Daily average actual and estimated reference evapotranspiration (ETo) values determined for the April–October period of 2021.

The black points on the graphs represent the actual ETo values calculated using the FAO-56 PM equation. The blue and red points represent the ETo values estimated using the original and modified equations of the models, respectively.

Figure 8 Daily average actual and estimated reference evapotranspiration (ETo) values determined for the April–October period of 2022.

The black points on the graphs represent the actual ETo values calculated using the FAO-56 PM equation. The blue and red points represent the ETo values estimated using the original and modified equations of the models, respectively.

The performances of the original equations for 2021 and 2022 were similar to those in the estimations made using long-term (2012–2020) climate data. The daily ETo values with the lowest and highest errors were estimated by Mahringer and Penman, respectively. The 2-year average MAE, MAPE, RMSE, and NSE values of Mahringer, which has the best-estimating performance, were realised as 0.90 mm day−1, 19.32%, 1.23 mm day−1, and 0.36, respectively. The same errors were determined as 2.43 mm day−1, 47.00%, 3.23 mm day−1, and −3.59 for the Penman, which has the worst-estimating performance. The models showing the nearest performance to Mahringer were WMO and Romanenko, respectively. The 2-year average MAE, MAPE, RMSE, and NSE values of WMO were realised as 0.86 mm day−1, 21.64%, 1.09 mm day−1, and 0.49, respectively. The same errors were determined as 0.95 mm day−1, 22.15%, 1.14 mm day−1, and 0.44 for Romanenko, respectively. The original equations of the Penman, Dalton, and Rohwer showed lower performance compared to Romanenko, Mahringer and WMO. The accuracy ranking of the original equations from best to worst according to their performance for daily average ETo estimates is as follows: Mahringer > WMO > Romanenko > Dalton > Rohwer > Penman. The daily ETo values estimated by Mahringer have a good level of accuracy (MAPE = 10–20%), while the daily ETo values estimated by other models have reasonable accuracy (MAPE = 20–50%). According to the NSE performances, Romanenko, WMO and Mahringer have a satisfying level of accuracy (NSE = 0.36–0.75), while Dalton, Rohwer and Penman have less satisfactory accuracy (NSE < 0.36) (Table 8).

Table 8 Performances of the original models in estimating daily ETo (2021–2022).

Models	MAE (mm day−1)	MAPE (%)	RMSE (mm day−1)	NSE	
2021	2022	Avg.	2021	2022	Avg.	2021	2022	Avg.	2021	2022	Avg.	
Dalton	1.25	1.88	1.57	27.82	34.99	31.41	1.64	2.39	2.02	−0.35	−1.22	−0.79	
Rohwer	1.37	2.05	1.71	29.78	37.76	33.77	1.82	2.65	2.24	−0.66	−1.72	−1.19	
Penman	1.96	2.90	2.43	41.04	52.96	47.00	2.67	3.79	3.23	−2.60	−4.57	−3.59	
Romanenko	0.80	1.09	0.95	20.95	23.35	22.15	1.01	1.27	1.14	0.49	0.38	0.44	
WMO	0.90	0.82	0.86	24.06	19.22	21.64	1.05	1.12	1.09	0.45	0.52	0.49	
Mahringer	0.75	1.05	0.90	18.13	20.51	19.32	0.99	1.47	1.23	0.52	0.19	0.36	
Note:

Mean absolute error (MAE), mean absolute percentage error (MAPE), root mean square error (RMSE) and, Nash–Sutcliffe Efficiency (NSE) express the deviation between the daily average actual ETo values calculated using the FAO-56 PM equation and the daily average ETo values estimated using the original equations of the Dalton, Rohwer, Penman, Romanenko, WMO and Mahringer models.

The modified equations demonstrated very similar performance in daily average ETo estimations using the 2021 and 2022 current climate data. The MAE, MAPE, RMSE, and NSE values of the daily average ETo values estimated for 2021 and 2022 using the modified equations varied between 0.34–0.50 mm day−1, 8.56–13.16%, 0.44–0.66 mm day−1, and 0.84–0.92, respectively. The daily ETo values with the lowest errors were estimated using the modified Dalton equation in both 2021 and 2022, similar to the estimations made using long-term climate data (2012–2020). The daily average ETo values with the highest errors were estimated with the modified Romanenko equation. The 2-year average MAE, MAPE, RMSE, and NSE values of the Dalton, which has the best-estimating performance, were determined as 0.35 mm day−1, 9.07%, 0.44 mm day−1, and 0.92, respectively. The same errors were calculated as 0.47 mm day−1, 11.92%, 0.58 mm day−1, and 0.85 for the Romanenko, which has the worst-estimating performance. The accuracy ranking of the modified equations from best to worst according to their performance for daily average ETo estimates is as follows: Dalton > Rohwer > Mahringer > Penman > WMO > Romanenko. The daily ETo values estimated by Dalton, Rohwer, and Mahringer demonstrate an excellent level of accuracy (MAPE < 10%), while those estimated by Penman, WMO, and Romanenko show a good level of accuracy (MAPE = 10–20%). The daily ETo values estimated by modified equations have a excellent level of accuracy (NSE > 0.75) according to their NSE performances (Table 9).

Table 9 Performances of the modified models in estimating daily ETo (2021–2022).

Models	MAE (mm day−1)	MAPE (%)	RMSE (mm day−1)	NSE	
2021	2022	Avg.	2021	2022	Avg.	2021	2022	Avg.	2021	2022	Avg.	
Dalton	0.35	0.34	0.35	9.58	8.56	9.07	0.44	0.44	0.44	0.90	0.93	0.92	
Rohwer	0.37	0.36	0.37	9.79	8.88	9.34	0.45	0.48	0.47	0.90	0.91	0.91	
Penman	0.42	0.43	0.43	10.95	10.07	10.51	0.52	0.60	0.56	0.86	0.86	0.86	
Romanenko	0.50	0.43	0.47	13.16	10.67	11.92	0.62	0.53	0.58	0.81	0.89	0.85	
WMO	0.45	0.46	0.46	11.65	10.66	11.16	0.57	0.66	0.62	0.84	0.83	0.84	
Mahringer	0.37	0.38	0.38	9.87	9.16	9.52	0.46	0.50	0.48	0.89	0.90	0.90	
Note:

Mean absolute error (MAE), mean absolute percentage error (MAPE), root mean square error (RMSE) and, Nash–Sutcliffe Efficiency (NSE) express the deviation between the daily average actual ETo values calculated using the FAO-56 PM equation and the daily average ETo values estimated using the original equations of the Dalton, Rohwer, Penman, Romanenko, WMO and Mahringer models.

According to the results of the calibration processes, the model with the most increased estimation performance in 2021 and 2022 was Penman. The performance of this model, which has the worst-estimating performance with an average accuracy of 53.00% (MAPE = 47.00%) in daily average ETo estimations made with the original equations, increased by 68.85% to 89.49% (MAPE = 10.51%) after calibration. The MAE (2.43 mm day−1), MAPE (47.00%) and RMSE (3.23 mm day−1) errors of ETo values estimated using the original equation of the Penman decreased by 82.31%, 77.64% and 82.66% to 0.43 mm day−1, 10.51% and 0.56 mm day−1, respectively. The performance of the Mahringer, which has the best-estimating performance with an accuracy of 80.68% (MAPE = 19.32%) in daily average ETo estimations made with the original equations, increased by 12.15% to 90.48% (MAPE = 9.52%) after calibration. The MAE (0.90 mm day−1), MAPE (19.32%) and RMSE (1.23 mm day−1) errors of ETo values estimated using the original equation of the Mahringer decreased by 57.78%, 50.73% and 60.98% to 0.38 mm day−1, 9.52% and 0.48 mm day−1, respectively. Using the original equations of WMO, Romanenko, Dalton and Rohwer, the daily average ETo values with an accuracy of 78.36% (MAPE = 21.64%), 77.85% (MAPE = 22.15%), 68.59% (MAPE = 31.41%), and 66.23% (MAPE = 33.77%), respectively, were obtained in the estimations made using the 2021 and 2022 data. These accuracy rates increased by 13.37%, 13.14%, 32.57% and 36.89% after calibration, respectively. Using the modified equations of the same models, the ETo values with an accuracy of 88.84% (MAPE = 11.16%), 88.08% (MAPE = 11.92%), 90.93% (MAPE = 9.07%) and 90.66% (MAPE = 9.34%) were estimated, respectively. The Mahringer performed better than all of the original equations of mass transfer-based models. The 2-year average accuracy level of the daily average ETo values estimated using the original equation of the Mahringer model was 2.32%, 2.83%, 12.09%, 14.45%, and 27.18% higher than the ETo values estimated with the original equations of WMO, Romanenko, Dalton, Rohwer and Penman, respectively. The Dalton performed better than all of the modified equations of mass transfer-based models. The 2-year average accuracy level of the daily average ETo values estimated using the modified equation of the Dalton model was 0.27%, 0.45%, 1.44%, 2.04%, and 2.85% higher than the ETo values estimated with the original equations of Rohwer, Mahringer, Penman, WMO, Romanenko, respectively (Fig. 9). The R2 coefficients ranging from 0.81 to 0.94 in 2021 and 2022 were obtained as an indicator of the correlation between ETo values estimated using mass transfer-based models and actual ETo values. Using the original equations, the R2 coefficient for the Mahringer, which has the best-estimating performance in the tests performed with both long-term (2012–2020) and 2021 and 2022 data, was determined as 0.92. Similarly, the R2 coefficient for the Dalton, which has the best-estimating performance in the estimations made with modified equations using the same data, was obtained as 0.92.

Figure 9 Mean absolute error (MAE), mean absolute percentage error (MAPE) and root mean square error (RMSE) values of mass transfer-based models for 2021 and 2022.

The blue and red bars on the graphs represent the MAE, MAPE and RMSE errors determined for the original and modified equations of the models, respectively.

Discussion

The Mahringer model, developed as part of studies aimed at estimating the evaporation realised from Neusiedl Lake in Austria, has been the best-performing model under the Van Lake microclimate conditions. The performance nearest to this model was showed by Romanenko. These models, which have a non-linear form, have performed better than other linear models (WMO, Dalton, Rohwer, and Penman) in daily average ETo estimations made with the original equations of the six mass transfer-based models. Similarly, the results obtained from many previous studies have shown that non-linear ETo estimation models perform better than linear models (Singh & Xu, 1997; Tabari, Grismer & Trajkovic, 2013; Djaman et al., 2017; Proutsos et al., 2023).

The monthly average actual ETo values determined using FAO-56 PM equation for April, May, June, July, August, September and October (2012–2020) were obtained as 2.49, 3.57, 4.91, 5.87, 5.58, 4.36 and 2.56 mm day−1, respectively. The nearest estimates to these values were made with the Mahringer model in the original equations and the Dalton model in the modified equations. Using the original equation of Mahringer, the monthly average ETo values estimated for the same months were determined as 1.75, 2.83, 4.78, 6.75, 6.76, 4.99 and 2.57 mm day−1, respectively. The monthly average ETo values estimated by the modified equation of Dalton were obtained as 2.51, 3.19, 4.46, 5.77, 5.82, 4.64 and 3.04 mm day−1, respectively. Similarly, in some studies conducted in Van conditions using FAO-56 PM, the following values were obtained for the same months, respectively: 3.04, 4.41, 5.57, 6.20, 5.65, 4.09, 2.29 mm day−1 (TAGEM, 2017); 2.91, 4.34, 5.67, 5.95, 5.63, 4.29, 2.60 mm day−1 (Usta et al., 2019); 3.03, 4.19, 5.45, 5.54, 5.48, 4.78, 2.96 mm day−1 (Saban Polu, 2021). There were significant differences between the ETo values determined from these studies using climate data measured before 2017 and the ETo values obtained from this study using climate data measured between 2012 and 2022. It is thought that the ETo differences are due to the drought experienced in the Eastern Anatolia Region between 2018 and 2022. According to the results of some studies conducted in recent years, it has been reported that there is a decreasing trend in precipitation and an increasing trend in temperature and evapotranspiration on the Van Lake closed basin, and this situation may increase the risk of drought in the basin in the coming years (Coşkun, 2020; Aydın & Öz, 2021).

Usta & Gençoğlan (2019) created empirical ETo estimation models based on air temperature and relative humidity compatible with the Van conditions. They obtained MAE errors ranging between 0.12–016 mm day−1 and MAPE errors ranging between 3.85–10.94%, respectively, as an indicator of the deviation between the daily ETo values estimated by these models based on multiple linear regression and the daily ETo values calculated using the FAO-56 PM equation. Similarly, Usta et al. (2019) and Uzunlar, Öz & Diş (2022) evaluated the temperature-based Blaney & Criddle and radiation-based Hargreaves & Samani empirical ETo estimation models in Van conditions and they created modified equations of these models compatible with local conditions. In these studies, where the FAO-56 PM equation was used as a comparison criterion, MAPE errors ranging from 9.00% to 32.00% and from 2.00% to 22.00% were obtained for the daily average ETo estimates made using the original equations of the Blaney & Criddle and Hargreaves & Samani models, respectively. In the daily average ETo estimates using the modified equations of the same models, MAPE errors decreased to values ranging from 6.30–18.00% and 0.041–8.00%, respectively. Similarly, in this study evaluating six mass transfer-based models, MAPE errors ranging from 15.86% to 33.68% were obtained for estimates made using the original equations. These errors decreased to values ranging from 7.51% to 8.89% after calibration.

Rashid Niaghi, Hassanijalilian & Shiri (2021) developed empirical ETo estimation models based on air temperature, radiation and mass transfer using spatial and temporal machine learning approaches. They reported that the radiation-based models (MAE = 0.36–0.61 mm day−1, RMSE = 0.55–0.77 mm day−1) illustrated the highest accuracy compared to mass transfer-based (MAE = 0.49–0.76 mm day−1, RMSE = 0.69–0.95 mm day−1) and temperature-based (MAE = 0.57–0.84 mm day−1, RMSE = 0.80–1.13 mm day−1) models. Similar to these mass transfer-based models that showed the nearest performances to the radiation-based models, the MAE and RMSE errors obtained for the Dalton, Rohwer, Mahringer, Penman, WMO and Romanenko models calibrated in compatible with the Van conditions varied between 0.29–0.36 mm day−1 and 0.33–0.44 mm day−1, respectively. The original equations of the Dalton, Rohwer, Penman, Romanenko and Mahringer overestimated ETo values by 23.26%, 24.63%, 33.68%, 16.47% and 15.86%, respectively. The modified equations of the same models overestimated ETo values by 7.51%, 7.65%, 8.42%, 8.86% and 7.85%, respectively. The original and modified equations of the WMO underestimated ETo values by 20.46% and 8.89%, respectively. Similar studies conducted in different climatic and environmental conditions, it has been confirmed that some mass transfer-based models overestimated ETo (Winter, Rosenberry & Sturrock, 1995; Djaman et al., 2015; Valipour, 2017). Tabari, Grismer & Trajkovic (2013) and Muhammad et al. (2019) reported that the WMO underestimated ETo.

Bormann (2011) revealed significant differences between the estimation performances of some mass transfer-based models evaluated in Germany conditions. Similarly, the performances of the original equations of the six mass transfer-based models evaluated in this study were higher in May, June, July, and August compared to their performances in April, September, and October. Using these equations, daily average ETo values with approximately 82.37% (MAPE = 17.63%) accuracy were estimated for May, June, July, and August, when monthly average solar radiation, air temperature, and wind velocity ranged between 17.04–24.53 MJ m−2 day−1, 13.40–22.90 °C, and 2.80–3.00 m s−1, respectively. In contrast, during April, September, and October, when monthly average solar radiation, air temperature, and wind velocity varied between 13.65–22.55 MJ m−2 day−1, 8.40–18.40 °C and 2.55–2.94 m s−1, respectively, the accuracy rate decreased to 77.34% (MAPE = 22.66%). Similarly, according to the results of studies realised in Iran conditions, Rezaei, Valipour & Valipour (2016) and Valipour, Gholami Sefidkouhi & Raeini-Sarjaz (2017) stated that the best weather conditions to use mass transfer-based models except Penman are 23.60–24.60 MJ m−2 day−1, 12.00–26.00 °C, and 2.50–3.25 m s−1 for solar radiation, air temperature, and wind velocity, respectively. They also expressed that Penman was the model with the worst-estimating performance in cities near the Caspian Sea and Persian Gulf. where the annual average relative humidity was higher than 65.00%. Similarly, the daily average ETo values with the lowest accuracy in Van on the coast of Van Lake were estimated by the original equation of the Penman. The annual average relative humidity in the city is 58.67%, and can reached 65.00% especially in April, May and October in coastal areas due to the increase in precipitation. In these months, monthly total precipitation increases to maximum levels with 54.90, 45.70 and 46.90 mm values, respectively. Monthly average relative humidity values are 61.76%, 57.01% and 54.93%, respectively. The original equation of the Romanenko model based on relative humidity and air temperature performed better than the other models in April, May and October when the relative humidity reached maximum levels. The best-performing models after Romanenko in these months were Mahringer and WMO, respectively. Similarly, Djaman et al. (2017) reported that Romanenko (MAE = 0.79 mm day−1), Mahringer (MAE = 1.11 mm day−1), WMO (MAE = 1.36 mm day−1), Rohwer (MAE = 1.42 mm day−1) and Dalton (MAE = 1.44 mm day−1) were the best-performing mass transfer-based models in coastal regions of Kenya. Proutsos et al. (2023) revealed that the best-performing models in the semi-arid Amoroussion and subhumid Heraklion cities of Greece were WMO (MAE = 0.69–0.91 mm day−1) and Mahringer (MAE = 0.73–0.83 mm day−1). They also declared that Rohwer (MAE = 1.14–1.76 mm day−1) and Dalton (MAE = 1.53–1.97 mm day−1) showed lower performance than WMO and Mahringer. Similarly, the original equations of Mahringer (MAE = 0.70 mm day−1), Romanenko (MAE = 0.79 mm day−1) and WMO (MAE = 0.79 mm day−1) performed better than Rohwer (MAE = 1.32 mm day−1) and Dalton (MAE = 1.24 mm day−1) in semi-arid Van conditions.

Irmak & Haman (2003) and Gundekar et al. (2008) stated that RMSE errors lower than 0.50 mm day−1 are acceptable for daily ETo values estimated using various empirical methods based on air temperature, radiation, or mass transfer (RMSE < 0.50 mm day−1). Similarly, Moratiel et al. (2020) declared that daily ETo values with MAE errors lower than 0.52 mm day−1 are acceptable in estimations made using empirical models (MAE < 0.52 mm day−1). In addition, NSE is one of the most widely used similarity measures in hydrology for calibration, model comparison and verification. It is stated that estimation models with NSE values above 0.75 have high levels of reliability, usability, and accuracy (NSE > 0.75) (Pushpalatha et al., 2012; Lufi, Ery & Rispiningtati, 2020; Duc & Sawada, 2023). When considering the acceptable error limits for RMSE (RMSE < 0.50 mm day−1) and MAE (MAE < 0.52 mm day−1), it is observed that none of the original equations of the Dalton, Rohwer, Penman, Romanenko, WMO, and Mahringer can be used in Van conditions. The RMSE (0.99–2.81 mm day−1) and MAE (0.80–2.14 mm day−1) errors obtained for the daily average ETo values estimated using these equations are above the acceptable error limit (Fig. 10). Similarly, the RMSE and MAE errors of the original equations of some mass transfer-based models evaluated under different climatic and environmental conditions were above the acceptable error limit (Tabari, Grismer & Trajkovic, 2013; Rácz, Nagy & Dobos, 2013; Djaman et al., 2016; Djaman et al., 2017; Proutsos et al., 2023). When considering the acceptable limit for NSE (NSE > 0.75), it has been concluded that the original equations of Mahringer (NSE = 0.81), WMO (NSE = 0.79), and Romanenko (NSE = 0.76) models can be used in Van conditions without the need for calibration, while the Dalton (NSE = 0.35), Rohwer (NSE = 0.25), and Penman (NSE = −0.49) models cannot be used (Fig. 10).

Figure 10 Accuracy and acceptability levels of six mass transfer-based models for the period of 2012–2022.

The black lines on the graphs represent the acceptable MAE (MAE < 0.52 mm day−1), MAPE (MAPE < 10.00%), RMSE (RMSE < 0.50 mm day−1), and NSE (NSE > 0.75) values. The blue and red lines represent the MAE, MAPE, RMSE and NSE values of the original and modified models, respectively.

The RMSE errors were obtained as 0.39, 0.41, 0.48, 0.51, 0.53, and 0.42 mm day−1, respectively, for the modified equations of Dalton, Rohwer, Penman, Romanenko, WMO, and Mahringer models in tests performed for the period of 2012–2022. These errors were below the acceptable error limit, except for Romanenko and WMO. The modified equations of Romanenko (RMSE = 0.51 mm day-1) and WMO (RMSE = 0.53 mm day-1) models performed above the acceptable error limit (RMSE > 0.50 mm day−1). These equations can estimate the daily average ETo with errors exceeding the acceptable error limit (RMSE < 0.50 mm day−1) by 2.00% and 6.00%, respectively. However, all of the MAE errors calculated for the modified equations remained below the acceptable error limit (MAE < 0.52 mm day−1). It has been concluded that daily average ETo values can be estimated with excellent accuracy (MAPE < 10%) using the modified models. The NSE values calculated for the modified equations of the six mass transfer-based models (NSE > 0.75) indicate that these equations can be used as alternatives to the standard FAO-56 PM equation in Van conditions. It has been observed that modified equations compatible with local conditions exhibit significantly higher accuracy compared to the original equations (Fig. 10). Similarly, several studies have demonstrated that the accuracy of some mass transfer-based models, calibrated by evaluating under different climatic and environmental conditions, has increased post-calibration (Zhai et al., 2010; Azhar & Perera, 2011; Djaman et al., 2016; Muniandy, Yusop & Askari, 2016; Rezaei, Valipour & Valipour, 2016; Valipour, Gholami Sefidkouhi & Raeini-Sarjaz, 2017; Celestin et al., 2020; Proutsos et al., 2023; Li et al., 2024).

Conclusions

This study was conducted to evaluate, calibrate (period: 2012–2020) and further validate (period: 2021–2022) six mass transfer-based reference evapotranspiration estimation models against standard FAO-56 PM model in Van Lake microclimate conditions of Turkey. The performances of these models, calibrated using the linear regression-based solver add-on of Microsoft Excel program were evaluated based on five statistical approaches (MAE, MAPE, RMSE, NSE, R2). The Mahringer model, developed to estimate evaporation from Neusiedl Lake in Austria was the best-performing model under the Van Lake microclimate conditions. The Romanenko model based on relative humidity and air temperature was the best performing model in April, May and October when the relative humidity reached maximum levels and had the greatest affect on ETo.

The disparities in capturing ETo dynamics among the six mass transfer-based models using the same independent variables (VPD, U2) are mainly caused by the models’ structure. The non-linear Mahringer and Romanenko models have performed better than the other linear models (WMO, Dalton, Rohwer, and Penman). The climate parameters that most affect ETo are known to be solar radiation, air temperature, relative humidity, wind velocity, and soil heat flux. Using the FAO-56 PM method, which utilizes all these parameters, ETo can be estimated with high accuracy. On the contrary, mass transfer-based models simplify the complicated relationship between ETo and VPD (an incarnation of RH) as a linear function and linear or square root function with wind velocity. Therefore, the estimation performances of these models with fewer parameters as independent variables are lower than the FAO-56 PM method. Another important issue affecting the estimation performance of mass transfer-based models is their compatibility with the climatic and environmental conditions of the region where they were developed. For this reason, they may exhibit lower performance under different conditions.

It has been observed that six mass transfer-based models are susceptible to changes in air temperature, wind velocity and relative humidity parameters and their accuracy has increased post-calibration. The validity of the daily average ETo values estimated by these models has been revealed through tests with the current climate data of 2021 and 2022, independent of the climate data for the 2012–2020 period used in the calibration. While the original models estimated ETo values with an accuracy ranging from 53.00% to 80.68% (MAPE = 19.32–47.00%), the accuracy increased to approximately 91.00% (MAPE = 9.00%) with the modified models. The results are likely to help minimize the error in ETo estimates within the context of irrigation schedules, water budgets, and drought action plans prepared under the Van Lake microclimate conditions.

It has been concluded that if all the data required for the standard FAO-56 PM method cannot be measured or supplied, the original Mahringer, WMO and Romanenko models, as well as the modified models created as a result of the calibration processes, can be used for daily average ETo estimates under local conditions, while the original Dalton, Rohwer and Penman models cannot be used. Additionally, the approach implemented in this study can be used in closed basin regions that have an altitude of over 1,500 m and similar topography and climatic conditions to those of the Van Lake closed basin. Finally, since the findings obtained from this study conducted for the Van Lake closed basin show that reference evapotranspiration tends to increase during the period 2012–2022 and that the basin may face a serious risk of drought in the coming years, it is recommended to take necessary precautions.

Supplemental Information

Supplemental Information 1 Calibrating of Dalton model.

Supplemental Information 2 Calibrating of Rohwer model.

Supplemental Information 3 Calibrating of Penman model.

Supplemental Information 4 Calibrating of Romanenko model.

Supplemental Information 5 Calibrating of WMO model.

Supplemental Information 6 Calibrating of Mahringer model.

Supplemental Information 7 Daily average ETo values estimated using the Dalton model (2012–2020).

Supplemental Information 8 Daily average ETo values estimated using the Mahringer model (2012–2020).

Supplemental Information 9 Daily average ETo values estimated using the Penman model (2012–2020).

Supplemental Information 10 Daily average ETo values estimated using the Rohwer model (2012–2020).

Supplemental Information 11 Daily average ETo values estimated using the WMO model (2012–2020).

Daily average ETo values estimated using the WMO model (2012–2020).

Supplemental Information 12 Daily average ETo values estimated using the Romanenko model (2012–2020).

Daily average ETo values estimated using the Romanenko model (2012–2020).

Supplemental Information 13 Testing of Dalton model (2012-2020).

Supplemental Information 14 Testing of Mahringer model (2012–2020).

Supplemental Information 15 Testing of Penman model (2012–2020).

Supplemental Information 16 Testing of Rohwer model (2012–2020).

Supplemental Information 17 Testing of Romanenko model (2012–2020).

Supplemental Information 18 Testing of WMO model (2012–2020).

Supplemental Information 19 Testing original equations (2021).

Supplemental Information 20 Testing original equations (2022).

Supplemental Information 21 Testing modified equations (2021).

Supplemental Information 22 Testing modified equations (2022).

Additional Information and Declarations

Competing Interests

Author Contributions

Data Availability

The author declares that he has no competing interests.

Selçuk Usta conceived and designed the experiments, performed the experiments, analyzed the data, prepared figures and/or tables, authored or reviewed drafts of the article, and approved the final draft.

The following information was supplied regarding data availability:

The calibration processes of the six mass transfer-based ETo estimation models; daily average actual and estimated ETo values determined for the period 2012–2020; the results of the tests performed for the original and modified equations of the six models using the data for the period 2012–2020; and the results of the tests performed for the original and modified equations of the six models using the current data of 2021 and 2022 are available in the Supplemental Files.

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
