# Peer review of "Comparison of the performances of six empirical mass transfer-based reference evapotranspiration estimation models in semi-arid conditions"

_PeerJ, doi:10.7717/peerj.18549_

## Round 0.1 · original submission · Major Revisions

Dear Dr. Usta

The reviewers have commented on your manuscript. You can find the attached reports. Based on the comments and suggestions of the expert reviewers, a major revision is needed for your article.

I request you check and correct the manuscript based on the reports.

Sincerely yours

·

Basic reporting

The manuscript illustrates alternative means of estimating evapotranspiration in a Turkish lake basin. The title is overly long and complex. It is recommended not to add the geographical location to the title, to underline the global relevance of the research, as required by an international journal. The terms "evaluation" and "calibration" are redundant because the calibration step is essential to refine the evaluation of evapotranspiration. The whole title should be far shorter and closer to the point, something like: "Comparing mass-transfer based estimates of evapotranspiration". The Introduction provides relevant background to support the choice of methods. It is clearly divided into several parts, each with its only subtitle. The paragraph referring to "(ETo) determined for the reference grass crop etc." is surprising because this method is not further developed. Therefore it should not be described in detail and it should be cited first in the Introduction. On the other hand, the FAO Method should be illustrated in full, by adding the formula to the Introduction, better highlighting what parameter is considered hard to establish to ensure its widespread use (as argued in Line 77). It is recommended that the any description of the Lake Basin be left to the Methods section. This should help in producing a more focussed Introduction.

Experimental design

The experimental design is relevant and original; this contribution is within the scope of the Journal. When introducing Lake Van, it should be stated clearly that this site was selected also because it offers the opportunity to produce an estimate of evapotranspiration using the FAO equation next to the other approaches, and therefore this allows a comparison between different methods.
The experimental design is incomplete, as it does not provide information about the data distribution and whether the data meet the statistical assumptions implicit in the error indicators used to compare the different approaches. Typically, this type of data is not normally distributed, yet the method proposed uses regression analysis. A more in-depth statistical analysis is required to justify the validity of this approach.
Some restructuring of the results section is necessary. The first paragraph of the Results is a description of Methods and it should be transferred to the former section despite the fact that it includes Figure 3 (methodological example). The first mention of "daily average ETo" should be accompanied by reference to the Figure where the data are illustrated.

Validity of the findings

Results are difficult to read and should be subdivided into sub-paragraphs each with own sub-title (daily averages, monthly averages, implementation of performance criteria, etc.), to better accompany the reader and facilitate the understanding of the articulation of the different methodological steps.
Please explain in greater detail HOW was the calibration performed, i.e. what coefficients were modifiede to adjust the estimated values to the real data.
Please discuss the statistical limitations of your approach, not least the potential autocorrelation of the different evapotranspiration estimates, based on very similar data. Can these be compared as if they were independent?
The comparison between different methods should be accompanied by a more inquisitive comment about the reasons that could explain the better performance of some estimates in relation to others, also by taking into account the statistical significance of the differential performance. By the way, the observation that "coastal regions of Kenya where dry and arid climate prevails" sounds rather strange given that coastal areas in Kenya are under the influence of marine moisture, characterised by a dual rainy season regime, and cannot be considered arid as some of the northern interior of the country (please check this statement).

Additional comments

The Conclusion is mainly a summary and not a critical Conclusion providing an uplifting outlook and highlighting the main results of the study.
The limitations of this study are rather obvious but not clearly stated by the authors: in the case of Van Lake, the natural equations would yield results that are grossly different from the most likely estimate produced by the FAO equation. Following "calibration" these equation become far closer. However, nothing guarantees that these equations could be used in another location or even in Lake Van for estimating evapotranspiration during a different time period. Please discuss similar limitations in your Conclusions and try t define a way forward to over come them.
Language should be improved by possibly engaging an English-mother tongue hydrologist.

Reviewer 2 ·

Basic reporting

no comment

Experimental design

no comment

Validity of the findings

no comment

Additional comments

In this study, the author has modified conventional ETo estimation equations using data from a single station. The results are presented using repetitive figures. In addition, the equations used in the study are relatively outdated as more advanced versions are now available. Typically, the literature indicates that data from a limited geographic region are used to derive these equations, and it is recommended that users adapt them for their specific regions. As such, the study does not make any new contributions. The graphs included all contain redundant information. The modification method used is not clearly defined. The study should be revised to better highlight potential innovations and to present the results in a more concise manner.
1. The measurement with a lysimeter corresponds to ETa, not ETo. Additionally, the Eddy Covariance systems are currently the most commonly used method for measuring ETa. This information should be added to the introduction.
2. In regions where ETo values vary within a wide range, an RMSE value below 1 can be considered acceptable. However, in regions where ETo values vary within a narrow range, an RMSE value below 0.1 can be deemed acceptable. The key point here is to use metrics that measure the overall success of the model, such as R, R2, NSE, and KGE. RMSE alone is reliable only when comparing the results of two models. Therefore, this statement needs to be revised.
3. The abstract does not mention which statistical method was used to modify the equations.
4. The statements between lines 160-172 should be included under the 'Study Area' section.
5. There is no need to redefine the study area in the 'Main Objective' section. Additionally, the method used for the modification process should be mentioned in this section.
6. The study mentions that values between April and October were analyzed, but it does not specify which crop is cultivated in the region during this period. This information should be included.
7. Working with data from a single station can be misleading. The study should incorporate data from multiple stations within the region. Data from a single station only reflects the conditions of the immediate surroundings and does not provide a comprehensive view of the Van region.
8. In the manuscript, the equations start with Equation 7. However, references to equations like Eq1 and Eq2 are present in the text, but these equations are not visible in the manuscript.
9. What method is being referred to when mentioning the Microsoft Excel program solver? This method needs to be specified in the study.
10. After Equation 12, the manuscript jumps to Equation 19. These sections should be corrected.
11. In the equations for MAE, MAPE, and RMSE, "i:1" should be written as "i=1."
12. Why was the average of the first 9 years taken to represent Figure 2, while the last two years are plotted separately?
13. Figure 3 is not suitable for inclusion in an academic paper. Sharing the tool used in Excel is unnecessary. Instead, the statistical method used should be emphasized.
14. As seen in Figure 6, the use of R2 has some drawbacks. Methods such as NSE or KGE should be utilized to test the performance of similar models.
15. Why were the predictions for 2021 and 2022 presented in separate graphs? They could also be shown on the same graph.

---

## Round 0.2 · Minor Revisions

Dear Dr. Usta

The reviewers have commented on your manuscript. You can find the attached reports. Based on the comments and suggestions of the expert reviewers, a minor revision is needed for your article.

I request you check and correct the manuscript based on the reports.

Sincerely

·

Basic reporting

This revision represents an improvement in relation to the previous submission. However, several minor mistakes persist and the manuscript would benefit from a language revision. It is recommended that it could be edited by a English mother-tongued hydrologist.

Experimental design

The experimental design seems sound, however, I had previously requested the author to check the coherence of his assumptions concerning data distributions. Are the data provided by the metereological station distributed in suach a way that the statistical modelling applied does not cause systematic bias? In particular, data that are not normally distributed can be assessed using the train of tests that is being proposed here and specifically the linear models? Should the data not be transformed beforehand? Could the lack of data transformation impact on the poor performance of some models?

Validity of the findings

The author could better clarifythe confidence he attributes to the data in a context of rapid climate change. It sems that despite this effort, new estimates could vary widely because of the current trends in temperature increase. The author should try to better frame the validity of the results under the current climate change scenario. Could data before the 1980s, when climate change was not perceptible, be compared to a recent period (climate change-affected) -say 2014-2024- to check how does model perfomarce react to it?

Additional comments

Instead of just stating which model achieves better performace, the author could try and reflect on the parameters/variables contained in the models that are more vulnerable to unexplaned variability or that could cause models to produce errors. Why do some models work better than others?
Could model performace dependent on the ability to measure certain parameters?
Could this discussion be broadened a little in the Conclusions to provide a feed for future research?

Reviewer 2 ·

Basic reporting

The author has made the necessary revisions. The manuscript is acceptable for publication.

Experimental design

-

Validity of the findings

-

Additional comments

-

---

## Round 0.3 · accepted · Accept

Dear Dr. Usta

I thank you for making the corrections and changes requested by the reviewers. I read and checked your valuable article carefully and am happy to inform you that the article has been accepted for publication in PeerJ.

·

Basic reporting

Clearly written.
Very sadly the title read "in semi-arid conditions", while "under semi-arid conditions" would be correct. I leave to the Editors to decide if this minor correction can still be implemented.

Experimental design

Originality acceptable (not an entirely new method but original in its implementation in the specific study reagion).
Well-defined, relevant and thoroughly explained.
Rigorous, after addressing concerns about data distributions.
Detailed explanation.

Validity of the findings

Relevant findings, discussed against sufficient literature evidence.
Data controlled and sound stats.
Conclusions support results.

Additional comments

worth publishing